# Seasonal and Nutritional Fluctuations in the mRNA Levels of the Short Form of the Leptin Receptor (*LRa*) in the Hypothalamus and Anterior Pituitary in Resistin-Treated Sheep

**DOI:** 10.3390/ani11082451

**Published:** 2021-08-20

**Authors:** Weronika Biernat, Malgorzata Szczęsna, Katarzyna Kirsz, Dorota Anna Zieba

**Affiliations:** Department of Animal Nutrition and Biotechnology, and Fisheries, Faculty of Animal Sciences, University of Agriculture in Krakow, 31-120 Krakow, Poland; weronikaa.biernat@gmail.com (W.B.); malgorzta.szczesna@urk.edu.pl (M.S.); katarzyna.kirsz@urk.edu.pl (K.K.)

**Keywords:** leptin, leptin receptor, nutrition, photoperiod, resistin, sheep

## Abstract

**Simple Summary:**

Research since the discovery of leptin has mainly focused on the long form of the leptin receptor. Currently, experiments on the short form of the leptin receptor have confirmed that not only is short form of leptin receptor present in the hypothalamus, but also expanded knowledge with information documenting the specific expression of that form of leptin receptor in selected areas of the hypothalamus and in the pituitary gland. In addition, we have shown that short form of leptin receptor expression levels are affected by day length, adiposity and resistin in sheep.

**Abstract:**

The short form of the leptin receptor (*LRa*) plays a key role in the transport of leptin to the central nervous system (CNS). Here, the resistin (RSTN)-mediated expression of *LRa* in the preoptic area (POA), ventromedial and dorsomedial nuclei (VMH/DMH),arcuate nucleus (ARC) and the anterior pituitary gland (AP)was analyzed considering the photoperiodic (experiment 1) and nutritional status (experiment 2) of ewes. In experiment 1, 30 sheep were fed normally and received one injection of saline or two doses of RSTN one hour prior to euthanasia. RSTN increased *LRa* expression mainly in the ARC and AP during long days (LD) and only in the AP during short days (SD). In experiment 2, an altered diet for 5 months created lean or fat sheep. Twenty sheep were divided into four groups: the lean and fat groups were given saline, while the lean-R and fat-R groups received RSTN one hour prior to euthanasia. Changes in adiposity influenced the effect of RSTN on *LRa* mRNA transcript levels in the POA, ARC and AP and without detection of *LRa* in the VMH/DMH. Overall, both photoperiodic and nutritional signals influence the effects of RSTN on leptin transport to the CNS and are involved in the adaptive/pathological phenomenon of leptin resistance in sheep.

## 1. Introduction

Domestication has led to an almost complete loss of seasonality in cattle, but it is retained in most breeds of sheep or goats [1]. At temperate latitudes, the reproductive activity of sheep is controlled mainly by the photoperiod and nutritional status of the organisms. The presence of sufficient energy reserves is critical to achieve successful reproduction. Adipocyte-derived leptin and resistin (RSTN) are major peripheral signals that control body fat reserves and the synthesis and release of many metabolic and reproductive hormones [2,3,4]. Our laboratory has focused on a physiological phenomenon that allows sheep to adapt to environmental changes—hypothalamic leptin resistance. Moreover, our most recent data show that RSTN is involved in this photoperiod and nutrition-dependent central leptin insensitivity in sheep which in that species occurs only during long day (LD) season [5,6]. Leptin affects target cells through its membrane receptor (LR). Six isoforms of LRs have been described thus far (LRa-f), the most important of which is the long isoform—*LRb*—because only this form is able to activate the main pathway of cell leptin signaling—JAK2-STAT3 [7]. An inhibitor of this leptin signaling pathway in the cell is suppressor cytokine signaling 3 (SOCS3), which inhibits leptin receptor phosphorylation [8]. *LRb* biosynthesis occurs in the hypothalamus, mainly in the arcuate nucleus (ARC), in neurons synthesizing proopiomelanocortin (POMC), neuropeptide Y (NPY) and agouti-related protein (AGRP) [4,5,9]. Leptin resistance occurs when, despite the high concentration of this anorectic hormone in the circulation, it has no effect on food intake. Loss of sensitivity to this adipokine may result from a mutation in the LR gene and lead to obesity. However, this form of resistance is very rare [10]. Commonly, the mechanisms underlying leptin resistance can be divided into two groups: those that interfere with leptin crossing the blood–brain barrier (BBB) and those associated with the action of this hormone on hypothalamic neurons. The causes of BBB-associated leptin resistance are mainly related to the transport of leptin through the choroid plexus, restriction of the access of this adipokine to neurons of the nervous system—saturation of the leptin transport system, which occurs at increased concentrations of this adipokine in the circulation or a reduction in the amount of leptin transporting proteins, mainly *LRa*, in capillary cell membranes [11]. In turn, the effect of leptin on hypothalamic neurons may be disturbed by the reduction or desensitization of *LRb* or by the inhibition of intracellular signaling pathways for this hormone, as previously mentioned [10]. However, there are also physiological situations in which natural resistance develops for leptin. This phenomenon is observed in species characterized by seasonal activity, e.g., sheep or Djungarian hamster, as well as during pregnancy, e.g., sheep or cattle [12,13,14].

In sheep, leptin must be transported by the BBB to activate first-order *LRb*-expressing neurons that are linked to downstream neurons to ultimately control food intake and energy expenditure [15]. The hypothalamic regulation of energy homeostasis involves an interconnected neural network that contains specialized neurons located in the ARC, the ventromedial nucleus (VMH), and the dorsomedial nucleus (DMH) [16,17]. In sheep, the most important neurons for reproductive processes are located in the ARC and the preoptic area (POA). To enter the brain and those neurons (ARC, VMH/DMH, POA), circulating leptin needs *LRa* to be transferred into the CNS. *LRa* is found in many organs and is not thought to play a definite role in signaling, but its participation in leptin transport is crucial. Thus, if the leptin concentration in the blood plasma is excessively high, it may cause saturation of *LRa*, which may further reduce the ratio of leptin transport across the BBB, ultimately leading to leptin resistance.

The aim of the current study was to determine the mRNA expression levels of the short form of the leptin receptor (*LRa*) in the hypothalamus and anterior pituitary (AP) of sheep. Information about this LR isoform is scarce in the literature, although it plays an important role in leptin signal into the brain. Taking into account the research model—seasonally breeding sheep—we investigated the role of photoperiod, nutritional status and RSTN, which also enters the brain via the BBB due to its low weight of 12.5 kD and is involved in the interaction between leptin resistance and increased expression of *SOCS3* [5,18].

Considering the aforementioned factors, the research was divided into two experiments; since sheep are seasonal animals, in the first experiment, we investigated the effect of the length of the day, the photoperiod, and the research was carried out during the long and short photoperiods. The second experiment was conducted during the LD period and highlighted the role of adiposity in the process of leptin resistance, since sheep are leptin resistant during the long day photoperiod. These factors were studied in the context of the expression of the short form of the leptin receptor, LRa, which is responsible for the transport of this adipokine to the brain.

## 2. Materials and Methods

The Second Local Ethics Committee on Animal Testing in Krakow, Poland, approved all the procedures conducted on the animals during these experiments (Protocols No. 16/2016 and 109/2018).

### 2.1. Animals

The studies were carried out at the Experiment Station of the Department of Animal Nutrition and Biotechnology, and Fisheries at the University of Agriculture in Krakow (longitude: 19°57 E, latitude: 50°04 N). A total of 50 female Polish Longwool sheep, a breed that exhibits strong reproductive seasonality, were prepared for the experiments (ovariectomized with estrogen replacement) as described in detail by Biernat et al. [4] to eliminate the variability resulting from the differences in the concentration of sex hormones and were implanted subcutaneously with a silastic estradiol implant. The mean circulating level of estradiol was 3.6 ± 0.3 pg/mL. Animals 2–3 years old were weighed at the start of the experiment (62 ± 3 kg) and were housed under natural photoperiodic and thermoperiodic conditions. The mean body condition score (BCS) was 3.1 ± 0.3 on a 1–5 scale (1 = exhausted; 5 = obese) [19].

### 2.2. Experimental Design

In both seasons, during short days (SD) and LD, sheep were often placed in wooden carts, in accordance with a previous report, to remove stress during the experiment [20]. The carts allowed the animals to stand or lie down freely while sampling and to make eye contact with other sheep. On the morning of each experiment, the sheep were cannulated with jugular vein catheters (Central and Peripheral Venous Catheters, Careflow^TM^, Argon, Billmed, Warsaw, Poland) for intensive blood sampling collection to determine estradiol, leptin and RSTN concentrations prior to administration of recombinant bovine resistin (rbresistin; CliniSciences, Nanterre, France) to determine the pretreatment status. Blood samples (5 mL) were collected at 10-min intervals for 4 h. On the second day, at the beginning of the experiment (time 0), blood samples were collected once through the catheter, and rbresistin was then injected through the same catheter. After each blood sample collection, the samples were dispensed into tubes containing 150 µL of a solution containing heparin (10,000 IU/mL) and 5% (*w*/*v*) ethylenediaminetetraacetic acid (EDTA) and placed on ice immediately. Plasma was separated by centrifugation and stored at −20 °C until analyses.

### 2.3. Animal Treatments

#### 2.3.1. Experiment 1. The Effects of Photoperiod and RbresistinTreatment on the Expression of *LRa* in Selected Brain Areas and Hormone Concentrations

A total of 30 ewes were fed twice a day with 07:00 and 16:00 diets to provide 100% of the National Research Institute of Animal Production recommendations for maintenance [21], with constant access to water. In both photoperiodic seasons, LD in May and SD in November, ewes (*n* = 15/season) were randomly assigned to one of the treatment groups (*n* = 5/group/season). The experimental groups were as follows: (1) control, administered saline (*n* = 10); (2) R1, administered a low dose of rbresistin (1.0 µg/kg of body weight (BW); *n* = 10); (3) R2, administered a high dose of rbresistin (10.0 µg/kg BW; *n* = 10).The ewes were humanely euthanized by captive bolt stunning 1 h after saline/rbresistin infusion. The brains with the infundibulum remaining intact were rapidly removed from the skulls of all the ewes and frozen on dry ice. Samples of the AP, hypothalamic ARC, VMH/DMH, and POA were aseptically isolated from the ewes 10–15 min postmortem. The collection of brain tissues was described in detail in previous papers [18]. Isolated tissues were frozen immediately on dry ice for storage at −80 °C. The brain tissue samples were rinsed in phosphate-buffered saline (PBS; Laboratory of Vaccines, Lublin, Poland), snap-frozen in liquid nitrogen, and then stored at −80 °C until analysis.

#### 2.3.2. Experiment 2. The Effect of BW and Rbresistin Treatment on the Expression of *LRa* in Selected Brain Areas and Hormone Concentrations

A total of 20 animals were randomly assigned into two groups: food restricted (lean, *n* = 10) and a high-energy diet developed to increase BW (fat, *n* = 10), which was achieved with an altered diet over 5 months. When food restriction was applied for extended periods, the sheep were housed in groups, and the lean animals received approximately 400 g of pasture hay/day supplemented with straw *ad libitum* as a filler. At the beginning of experiment the average BCS was 3.1 ± 0.3. The objective of the adiposity score was to alter the BCS of animals to 2 [19]. The fat animals received pasture hay *ad libitum* plus a dietary supplement of approximately 1 kg lupin grain/animal/week, which increased adipose deposition. The animals were weighed every two weeks, and target weights were reached by 4 months, after which the diets were maintained for another month as described in a paper by Zieba et al. [5]. The nutrient requirements of the sheep and the exact compositions of the diets were determined based on Instytut National de la Recherche Agronomique (INRA) [22]. The day before the experiment, the sheep were assigned to one of four treatment groups (*n* = 5/group/treatment). The experimental groups were as follows: the lean (*n* = 5) and fat (*n* = 5) groups were administered saline (5.0 mL), and the lean-R (*n* = 5) and fat-R (*n* = 5) groups were intravenously administered one dose of 5.0 µg/kg BW (5.0 mL) rbresistin. On the day of the experiment, 1 h after saline/rbresistin infusion, the animals were humanely euthanized by captive bolt stunning. The brains were collected using the same methodology as described in Experiment 1. The brain tissue samples were rinsed in phosphate-buffered saline (PBS; Laboratory of Vaccines, Lublin, Poland), snap-frozen in liquid nitrogen, and then stored at −80 °C until real-time PCR analysis.

### 2.4. Hormone Assays

Estradiol concentrations were determined using commercially available enzyme immunoassay (EIA) kits (DRG Instruments GmbH, Marburg, Germany) according to the manufacturer’s instructions. The inter- and intraassay precision values exhibited CVs of 3.46% and 2.4%, respectively. Assay sensitivity was 1.9 pg/mL. RSTN concentrations were determined using commercially available EIA kits (Cloud-Clone Corp., Katy, TX, USA) according to the manufacturer’s instructions. The inter- and intraassay precision values exhibited CVs of 6.5% and 2.4%, respectively, and the assay sensitivity was 2.2 pg/mL. Plasma leptin concentrations were determined using commercially available RIA kits (Multi-species Leptin RIA, EMD Millipore Co., Billerica, MA, USA) according to the manufacturer’s instructions. The inter- and intraassay precision values exhibited CVs of 3.5% and 2.2%, respectively. Assay sensitivity was 0.8 ng/mL.

### 2.5. Statistics

#### 2.5.1. Statistical Analysis

All hormone data are presented as the mean ± SEM. Data analysis was performed by a series of two-way ANOVAs using SigmaPlot^®^ statistical software (version 11.0; Systat Software Inc., Richmond, CA, USA), preceded by Grubb’s test to identify outliers. All data sets with failed tests of normality and/or equal variance were transformed as natural logarithms. If the main effects or their interactions were significant, the Holm–Sidak test was used as a post-ANOVA test to compare individual means. A *p*-value < 0.05 was considered to indicate statistical significance.

#### 2.5.2. Molecular Analysis

The mRNA expression of *LRa* was measured using the real-time PCR method. Tissue homogenization was performed with a rotor-stator homogenizer (Omni TH, Omni International, Inc., Kennesaw, GA, USA) and single-use tips (Soft Tissue Omni Tip Plastic Homogenizing Probes, Omni International, Inc., Kennesaw, GA, USA). Total RNA was isolated using TRIzol reagent (Ambion Inc., Austin, TX, USA) following the manufacturer’s protocol. The samples were incubated at 42 °C for 2 min with gDNA Wipeout Buffer (QuantiTect Reverse Transcription Kit; Qiagen, Hilden, Germany) to eliminate contamination of genomic DNA. Subsequently, to obtain samples of cDNAs by reverse transcription, isolates of RNA (1.0 µg) were incubated with Quantiscript reverse transcriptase and RT primer mix (QuantiTect Reverse Transcription Kit; Qiagen, Hilden, Germany) at 42 °C for 15 min. The reaction was terminated by heating the samples to 94 °C for 3 min. Each cDNA was amplified in triplicate using an Applied Biosystems 7300 Real-Time PCR System, TaqMan Gene Expression Master Mix, specific primers (900 nM) corresponding to the target/reference genes (Sequence Detection Primers) and specific probes (250 nM)corresponding to the target/reference genes (TaqMan MGB Probes) supplied by Life Technologies (Foster City, CA, USA). The primers and probes were designed using Primer Express software v. 2.0 (Applied Biosystems; Foster City, CA, USA) and are described in Table 1. The thermal profile for real-time PCR was as follows: (1) 50 °C for 2 min—initial incubation; (2) 95 °C for 10 min—activation of polymerase; (3) 40 cycles for denaturation (95 °C for 15 s) and annealing/elongation (60 °C for 60 s). The collected data were recorded with Applied Biosystem 7300 Real-Time PCR System SDS software.

#### 2.5.3. Molecular Data Analysis

The expression levels were calculated using relative quantification (RQ) analysis, and the results are expressed as a function of the threshold cycle (Ct), which is a value corresponding to the fractional PCR cycle number at which the fluorescent signal reaches the detection threshold. The data were analyzed using the 2^−ΔΔCt^ method, and Ct values were converted to fold-change RQ values. The RQ values from each gene were used to compare target gene expression across all groups. The mean mRNA expression levels for the target genes in each sample were standardized against the expression of a reference gene (cyclophilin; *CPH*) and expressed relative to the calibrator sample. The variation in the Ct values for *CPH* among the treatment groups was not significant (*p* > 0.05).

For the first experiment, the mean ΔCt value for tissue collected from the control group was adopted as a calibrator to compare the changes in target gene expression between all the treatment groups in the indicated season. For the second experiment, the mean ΔCt value for the indicated tissues collected from the lean group was used as a calibrator to compare the changes in target gene expression among all the treatment groups.

Differences in the means were compared with SigmaPlot statistical software (version 11.0; Systat Software Inc., Richmond, CA, USA) using all pairwise multiple comparison procedures (Tukey test), preceded by the determination of a significant F-value. Differences were considered statistically significant when *p* < 0.05.

## 3. Results

### 3.1. The Effects of Photoperiod and Rbresistin Treatment on the Expression of LRa in Selected Brain Areas and Hormone Concentrations

#### 3.1.1. Expression of mRNA *LRa* in Selected Brain Areas

*LRa* transcripts were detected at varying levels in the examined hypothalamic tissues, namely, the POA and ARC; however, no detection was noted in the VMH/DMH during either the LD or SD seasons (Figure 1A,B). In the AP, during the LD season, the expression of *LRa* increased proportionally with the administered dose of rbresistin: 2-fold (*p* < 0.01) for the R1 group and 2.5-fold (*p* < 0.01) for the R2 group compared to the control group. A significant increase (*p* < 0.01) was also found between the treatment groups (Figure 1A). In the SD season, exogenous RSTN decreased (*p* < 0.001) the mRNA transcript level of *LRa* in the R1 group and R2 group (Figure 1B).

*LRa* expression was not detected in the ARC and POA during the SD season. During the LD season, the *LRa* transcripts in the POA decreased 3-fold (*p* < 0.001) in the R1 group and 2-fold (*p* < 0.01) in the R2 group compared to the control group (Figure 1B). During the LD season, within the ARC, the mRNA levels of *LRa* increased 4.5-fold after rbresistin injection at a low dose compared to the levels noted in the control (*p* < 0.01) and decreased 2-fold after injection of a high dose (*p* < 0.05) in the R2 group (Figure 1B).

#### 3.1.2. Estradiol Concentration

The mean circulating concentration of estradiol (mean ± SEM) was 3.6 ± 0.3 pg/mL.

#### 3.1.3. Resistin Concentrations

During the LD and SD seasons, there were no differences (*p* > 0.7) between the mean concentration of circulating RSTN (mean ± SEM) in the control groups of sheep (Figure 2).

During the LD season, the administration of 1.0 µg/kg BW rbresistin (*p* < 0.01) and that of 10.0 µg/kg BW rbresistin (*p* < 0.001) increased the concentration of circulating RSTN compared to those of the control groups. No difference was observed in circulating RSTN concentrations between the LD and SD seasons (*p* = 0.08) within the R1 groups. In the SD season, the mean circulating RSTN concentration was higher (*p* < 0.05) in the R1 group than in the control group, and the administration of a high dose of RSTN increased (*p* < 0.001) the RSTN level in the R2 group of sheep vs. the control group (Figure 2).

#### 3.1.4. Leptin Concentrations

The mean concentration of circulating leptin (mean ± SEM) within the control and all treatment groups was significantly lower (*p* < 0.001) during the SD season than during the LD season (Figure 3). 

During the LD season, rbresistin at a dose of 1.0 µg/kg BW increased the mean leptin concentration compared to that observed in the control group (*p* < 0.001). A dose of 10 µg/kg BW rbresistin elevated the circulating leptin concentration compared to that of the control group (*p* < 0.001) in the LD season. During the SD season, doses of 1 and 10 µg/kg BW rbresistin significantly increased the concentration of leptin (*p* < 0.001) compared to that of the Control group (Figure 3).

### 3.2. The Effect of BW and Rbresistin Treatment on the Expression of LRa in Selected Brain Areas and Hormone Concentrations

#### 3.2.1. Expression of mRNA *LRa* in Selected Brain Areas

*LRa* transcripts were detected at varying levels in the examined hypothalamic tissues: the POA and ARC in the non-treated lean group and RSTN-treated (lean-R and fat-R) groups of sheep, with the exception of non-detectable levels of expression in the POA, VMH/DMH and ARC of the fat groups of ewes (Figure 4).

No detection of *LRa* transcript levels was noted in the VMH/DMH for the lean, lean-R and fat-R groups. In the ARC, the expression of *LRa* was 3-fold higher (*p* < 0.001) in the lean-R group and 2-fold higher (*p* < 0.001) in the fat-R vs. lean group of ewes.

There were significant differences (*p* < 0.01) between the lean-R and fat-R ewes in terms of the ARC. The 4-fold and 5-fold decreases in mRNA *LRa* transcript levels were noted in the lean-R group compared to the lean and fat-R groups, respectively (Figure 4).

There was a significant (*p* < 0.001) change in the AP *LRa* transcript levels between the lean and fat-R groups (Figure 4).

#### 3.2.2. Resistin Concentrations

The injection of 5.0 µg/kg BW rbresistin increased (*p* < 0.05) circulating RSTN concentrations in the fat and lean-R sheep compared to the non-treated lean ewes (Figure 5).

Significant differences (*p* < 0.05) were noted between the fat and fat-R groups and between the lean-R and fat-R groups (*p* < 0.05).

#### 3.2.3. Leptin Concentrations

Circulating concentrations (mean ± SEM) of leptin after rbresistin treatment were higher (*p* < 0.01) in the fat and lean-R sheep than in the lean group. Recombinant bovine resistin increased (*p* < 0.001) endogenous leptin concentrations in the fat-R group vs. the fat and lean-R experimental groups of sheep (Figure 6).

## 4. Discussion

Studies concerning leptin functions carried out since 1994 demonstrated that *LRa* is the most highly characterized short form and is expressed at high levels in the kidney, lung, choroid plexus, brain microvessels and pituitary gland [23]. Few studies primarily on rodents have been performed on the presence of *LRa* in the brain [24,25]. The results of the current research confirmed the observations of Hileman et al. [26] that *LRa* is present in the hypothalamus and extended the information documenting specific hypothalamic nuclei expression. In addition, we showed that the levels of *LRa* expression are related to RSTN, adiposity and day length. The last factor, the photoperiod, plays a crucial role in every aspect of physiological processes in sheep [27].

Experiments conducted in the early 1990s showed that defects in the transport of leptin through the BBB played a major role in the pathogenesis of obesity. In 1997, two studies showed that animals with diet-related obesity went through a period in which they did not respond to peripherally administered leptin but still responded to centrally administered leptin directly into the brain [28,29]. This was evidence not only that leptin exceeded the BBB in ineffective amounts but also that there was a period in which resistance to the BBB was dominant to insensitivity to leptin at the receptor/postreceptor level (referred to as central resistance). Further research from some laboratories has shown impairment of pharmacokinetics in blood–brain transport in several rodent obesity models, including diet-induced obesity, obesity-prone rats, and Koletsky rats; the last two represent genetic models where brain leptin receptors are deficient or absent [30,31,32].

One of the hypotheses explaining leptin resistance syndrome in sheep is saturation of *LRa*, which reduces leptin transport from the bloodstream to the hypothalamus and is termed peripheral resistance. High concentrations of leptin noted in both sheep models—obese (as a pathological process) and individuals during LD season (as a physiological adaptation)—demonstrated in our study indicated that hypothalamic and pituitary expression of mRNA *LRa* transcripts is highly correlated with nutritional and photoperiodical signals.

Neuronal nuclei within the ovine hypothalamus are in effect multivalent chemosensors and integrators of information that control appetite and food intake, the burning of energy and the deposition of fat (three factors that collectively determine energy balance). Physiological control of BW requires the monitoring and integration of many signals that circulate in the body and provide information from the periphery (stomach, pancreas, fat stores, etc.) to the brain by a ligand to its receptors [33]. These key signals include not only circulating metabolites such as glucose, free fatty acids and amino acids but also specifically secreted hormones such as leptin, RSTN, ghrelin and insulin. Understanding how the networks of the hypothalamus respond to these signals and integrate them to alter their output is still a subject of research in animal models, especially as unique as sheep exhibiting circannual reversible central leptin insensitivity because the hypothalamus and pituitary gland of sheep are opposite to each other and are leptin-resistant for a strictly defined period of the year [14,34,35].

In seasonally breed sheep, physiological leptin concentrations are higher during the LD season than the SD season [20], the concentration of leptin in the blood plasma of sheep increases by 180% [34], but this change is not related to the anorectic effect of leptin [36]. In our studies, both animal models—LD sheep as a physiological example of leptin resistance and sheep with high BW and high adiposity resulting from diet-induced obesity—demonstrated high leptin concentrations. In those two models, highly elevated expression of *LRa* was observed in the ARC nuclei—the neurons that transmit the leptin signal first directly to the CNS. Administration of a relatively low dose of RSTN (1.0 or 5.0 µg per kg BW) significantly increased the mRNA transcripts of *LRa* in two experiments in groups of LD sheep (R1 group) fed *ad libitum* and lean-R sheep nutritionally restricted. No detection of *LRa* expression in diet-induced fat sheep in the POA, VMH/DMH, or ARC proved that leptin transport by LRa in that group of sheep was disturbed, as has been indicated for diet-obese rats and mice [26,37]. Moreover, in obese mice, reduced brain leptin uptake has been shown [31]. In contrast to those observations, in fasting mice, significant elevation of leptin uptake into the brain has been noticed with no differences in the expression of *LRa* vs. control animals [26]. In those studies, Hileman et al. [26] demonstrated that the short isoform of the leptin receptor is an active participant in the transfer of leptin between the blood and brain and and this process depends on adiposity. In a paper devoted entirely to short isoforms of the leptin receptor, Hilmans’ group considered factors involved in regulation of leptin transport other than leptin concentration. Undoubtedly, such a factor in seasonal animals is photoperiod [36,38]. The present data support the hypothesis that photoperiod regulates the expression of *LRa* and transport of leptin into the brain in sheep. These data supported experiments by Adam et al. [36], who studied the cerebrospinal fluid to plasma leptin concentration ratio during LD and SD seasons. Interestingly, the expression of *LRa* noted in the SD season in sheep only in the pituitary gland confirmed our earlier observation concerning leptin signaling component expression [20]. In those experiments, *SOCS3* expression in ovine pituitaries was dependent on leptin dose and season. Significant changes (increased expression) were noted only during the SD season. This is in complete contradiction with the pattern of expression of this factor in the ovine hypothalamus during an LD season [20]. However, this indicates an absolute different sensitivity of individual brain regions to the action of leptin and signal transmission and helps to explain the lack of expression of *LRa* in the hypothalamus area in an SD season and the expression that occurs in the pituitary area during that season. Our study is the first to directly compare *LRa* expression in lean, normal-weight and obese sheep in both the hypothalamus and pituitary gland in relation to RSTN effects. Experiments by Asterholm’s group showed that chronically high RSTN concentrations directly caused leptin resistance and increased *SOCS3* expression in rat hypothalami [39]. Zieba et al. [5] confirmed in sheep that RSTN can be added to a group of factors creating a leptin central insensitivity phenomenon. Presently, it is indicated that RSTN elevated the expression of *LRa* in lean and fat ewes (POA, ARC and AP), which we noted as well for sheep with different nutritional statuses for *LRb* in previous experiments [5], which suggests that this adipokine is involved in leptin translocation and leptin signaling to the most important hypothalamic nuclei responsible for energy resources (ARC) and for POA for reproduction. Although our findings clearly indicate that *LRa* expression in selected hypothalamic nuclei and AP are altered in animals with different adiposity and RSTN treatments, the mechanism(s) responsible for this is unclear. As the short leptin isoform seems to be significantly involved in mediating leptin uptake into the brain, we originally hypothesized that changes in *LRa* expression should be associated with RSTN and leptin resistance phenomena that occur in the LD season and are related to the amount of fat in sheep.

As mentioned at the beginning of the discussion, we did not observe the detection of *LRa* in the VMH/DMH in the second experiment and in the POA, VHM/VDH and ARC during the SD season in the first experiment. Therefore, we cannot rule out the possibility that different results might be obtained if different brain regions had been compared. Binding sites for some cytokines have been largely located in the brain. Additionally, it is possible that changes in short *LRa* mRNA expression occur in only certain brain regions. Regional variation, if it occurs, is suggested by the findings of regional differences in brain uptake of leptin [40,41,42,43,44,45,46]. Experiments on mice demonstrated that leptin receptors are also situated outside the BBB in the central region of the hypothalamic ARC [36,46]. Data from that experiment point out an important role for a specialized subgroup of astroglia in the highly selective uptake of molecules such as NPY from the circulation in the median eminence and adjacent ARC. If this is the case for rats, it is also possible for sheep.

Further investigation will be necessary to determine whether short leptin receptor isoforms and proteins are present in ovine astrocytes participating in the uptake of circulating molecules into these regions of the CNS.

Leptin transport into the brain appears to be a limiting step in the modulation of its central effects, and it participates in leptin resistance not only in rodents and humans but also in sheep. The development of treatment strategies to circumvent this problem may offer attractive opportunities for pharmaceutical intervention in the pathogenesis of obesity, which is the subject of our other study using MTS-leptin—a modified leptin with improved permeation across the BBB. MTS-leptin [47] has been identified and used as a protein capable of crossing the BBB independent of the leptin BBB transporter.

## 5. Conclusions

In seasonal animals, intense research has been conducted to provide explanations for the relationships between hormones engaged in the regulation of reproduction, energy homeostasis and metabolism and the processes controlled by these relationships. These studies have been conducted not only in theoretical but also in practical terms for the treatment of pathological phenomena associated with endocrine dysfunction in animals or related to the economic viability of farming. Due to their strictly regulated adaptation to environmental conditions related to the plasticity of their endocrine system, as well as the presence of physiological leptin resistance, sheep represent a particularly interesting model for such studies. We found that *LRa* mRNA was highly expressed in both hypothalamic nuclei engaged in energy balance and reproduction (ARC, POA) and the pituitary gland and was dependent on RSTN treatment, nutritional status and photoperiod. We further showed that *LRa* expression and RSTN are integral parts of leptin translocation and signaling. Thus, leptin resistance (peripheral and/or central) may be a result of RSTN- and photoperiod-dependent hyperleptinemia, adiposity, disturbance in leptin signaling through elevated SOCS3 expression and decreased transport of leptin with *LRa* from blood into the CNS in sheep.

## Figures and Tables

**Figure 1 animals-11-02451-f001:**
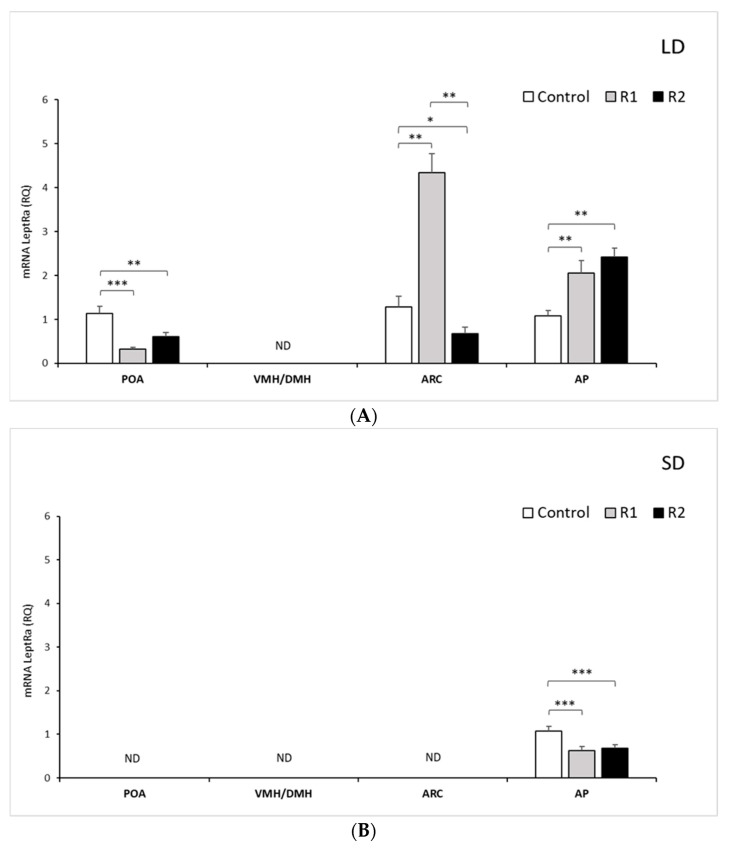
Leptin receptor expression.The mean expression (±SEM) of the short form of leptin receptor (*LRa*) mRNA in ovine preoptic area (POA), ventro- and dorsomedial nuclei (VMH/DMH), arcuate nucleus (ARC) and anterior pituitary (AP) collected during long-day (LD) (**A**) and short-day (SD) (**B**), photoperiods. The expression of *LRa* mRNA is reported in arbitrary units (RQ) relative to cyclophilin mRNA expression and expressed relative to the calibrator sample. * *p* < 0.5, ** *p* < 0.01 and *** *p* < 0.001 denote differences relative to the control or between the indicated group.

**Figure 2 animals-11-02451-f002:**
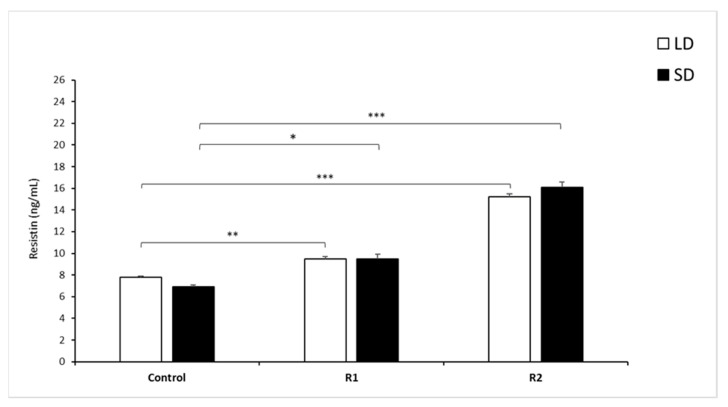
Plasma resistin concentrations. Mean concentrations of circulating (±SEM) resistin in saline (Control) and recombinant bovine resistin-treated groups (R1—low dose; R2—high dose of rbresistin). Only the most important significant differences are indicated by asterisks. Means without a common letter differ. * *p* < 0.05, ** *p* < 0.01, *** *p* < 0.001, denote difference between groups.

**Figure 3 animals-11-02451-f003:**
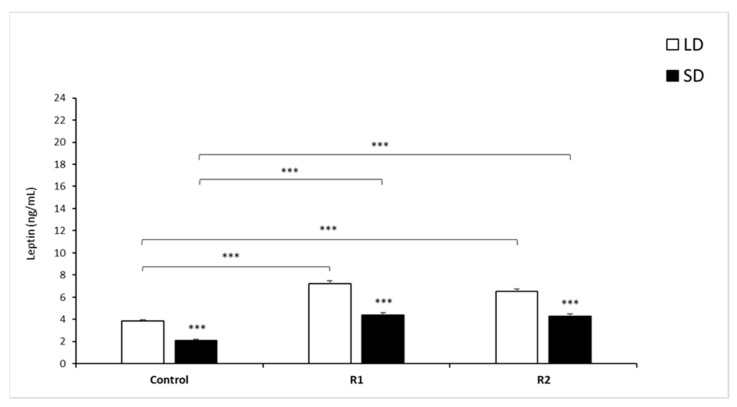
Plasma leptin concentrations. Mean concentrations of circulating (±SEM) of leptin in saline and recombinant bovine resistin-treated groups (R1—low dose; R2—high dose of rbresistin) during the long (LD) and the short (SD) photoperiods. Only the most important significant differences are indicated by asterisks. *** *p* < 0.001 denotes difference between groups.

**Figure 4 animals-11-02451-f004:**
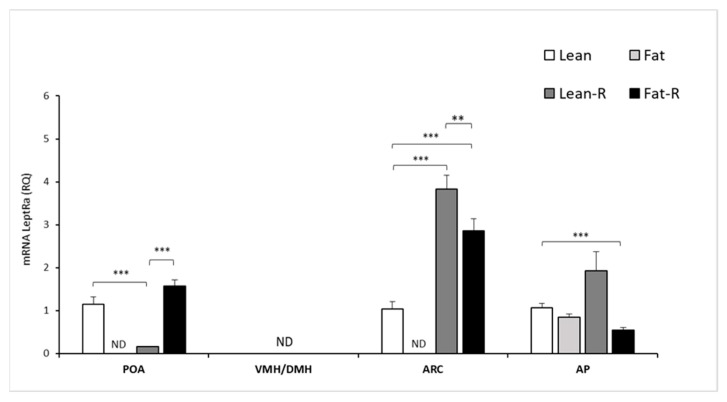
Leptin receptor isoform a mRNA expression. The mean mRNA expression (±SEM) of the short form of the leptin receptor (*LRa*) in the ovine preoptic area (POA), ventro- and dorsomedial nuclei (VMH/DMH), arcuate nucleus (ARC) and anterior pituitary gland (AP). The expression of *LRa* mRNA is reported in arbitrary units (RQ) relative to cyclophilin mRNA expression and expressed relative to the calibrator sample. Differences relative to the control or between the other groups are indicated with ** *p* < 0,01; *** *p* < 0.001. ND—non-detected.

**Figure 5 animals-11-02451-f005:**
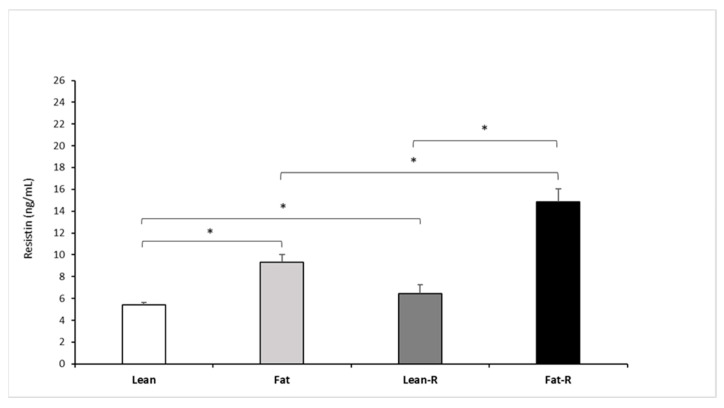
Plasma resistin concentrations. Mean concentrations of circulating (±SEM) of resistin in non-treated (Lean and Fat) animals and animals treated with recombinant bovine resistin (Lean-R and Fat-R) after 5 months of body weight alteration (*n* = 5 per group). * *p* < 0.05 denotes difference between groups.

**Figure 6 animals-11-02451-f006:**
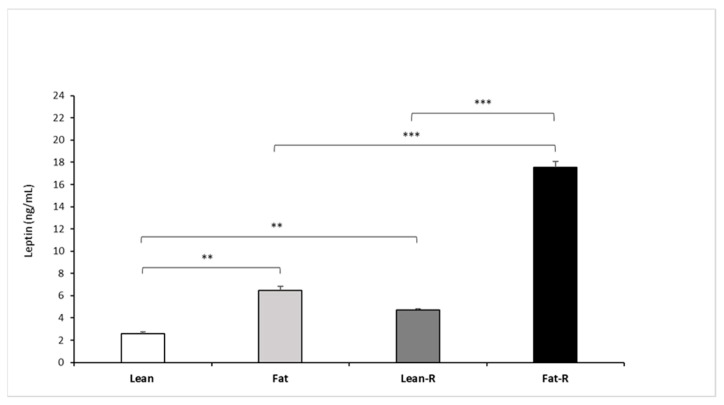
Plasma leptin concentrations. Mean concentrations of circulating (±SEM) of resistin in non-treated (Lean and Fat) animals and animals treated with recombinant bovine resistin (Lean-R and Fat-R) after 5 months of body weight alteration (*n* = 5 per group). ** *p* < 0.01, *** *p* < 0.001 denote difference between groups.

**Table 1 animals-11-02451-t001:** Sequences of oligonucleotides used as primers and probes to analyze the mRNA expression of cyclophilin (*CPH*; reference gene), the short form of the leptin receptor (*LRa*; target gene) in sheep.

Gene	PrimerSequence (5′–3′)	Probe Sequence (5′–3′)	Amplicon Size	Gen Bank Accession Number
***CPH***	CGGCTCCCAGTTCTTCATCA	FAM-CGTTCCGACTCCGC-MGB	64 bp	D14074
	ACTACGTGCTTCCCATCCAAA			
***LRa***	TCAAAGTATGTCCGTTCTCTTCTG	FAM-TGTTTTGGGAAGATGTTC-MGB	131 bp	NM_001009763.1
	TCTTATTGCTTGGAACATTGTCA			

## Data Availability

Data are available from the corresponding author under reasonable request.

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
