# Peer review of "Seasonal and Nutritional Fluctuations in the mRNA Levels of the Short Form of the Leptin Receptor (LRa) in the Hypothalamus and Anterior Pituitary in Resistin-Treated Sheep"

_animals, 2021, doi:10.3390/ani11082451_

Round 1

Reviewer 1 Report

The reviewed original paper concerns the short form of the leptin receptors and in particular its presence in the anterior pituitary and different regions of the brain in sheep. Authors also have studied effects of day length, adiposity and resistin of this receptor in sheep.

The major finding is that the short form of leptin receptor expression levels are affected by day length, adiposity and resistin in sheep, which add to the existing literature.

The study is well designed and methods are adequate explained. Improvement of the result section should be done. More information, especially on the limitation of the study should be added to the discussion section. The detailed revisions are listed below:

  1. 3. Please add space between “[4]” and “to” in the sentence: “(ovariectomized with estrogen replacement) as described in detail by Biernat et al. [4]to”
  2. 3. Please delate “mean ± SD “ after (62 ± 3 kg)
  3. 4. Please re-write the following sentence so it is understood “The objective was to alter the BCS of the animals to2 after Russel et al. [21].”
  4. 4. Authors wrote: “All hormone data are presented as the mean ± SEM. Data analysis was performed by a series of two-way ANOVAs using SigmaPlot® statistical software (version 11.0; Systat Software Inc., Richmond, CA, USA), preceded by Grubb’s test to identify outliers.” Could you please provide information how many outliers you had in this experiments ?
  5. 5 (bottom) “In the AP, during the LD season, the expression of LRa increased proportionally with the administered dose of rbresistin: 2-fold (p < 0.01) for the R1 group and 2.5-fold (p ≥ 0.01)”. Should be p<0.01.
  6. 6. “In the SD season, exogenous RSTN de-creased (p ≥ 0.001) the mRNA transcript level of LRa in the R1 group and R2 group (Figure 1b)” Should be p<0.001
  7. 6. Should be Figures 1A and B (plural) not “Figure 1A and B”.
  8. 6. Please reorganize the figure so data are presented from the rostral to caudal part of the brain. The data should be presented in the following direction: POA, VMH/DMH, ARC and lastly AP. The same rule applies to Fig. 4.
  9. 6 In the sentence “LRa expression was not detected in the ARC or POA during the SD season” please replace “or” with “and”
  10. 7 “…the LRa transcripts in the POA decreased 3-fold (p <0.001)in the R1 group and 2-fold”, please inset a space between “(p <0.001)” and “in”
  11. Figure 2 – Please indicate only the most significant and important statistical differences which are disused in the results. Please add this information to the figure legend “only the most important significant differences are indicated by asterisks”.
  12. 7. “During the LD season, the administration of 1.0 μg/kg BW rbresistin(p < 0.01)”. Please insert a space between “rbresistin” and “(p < 0.01)”.
  13. 7. Please delate “Means without a common letter differ” from the Fig. 3 legend as there is no letter on this fig. Please indicate only the most significant and important statistical differences which are disused in the results. Please add this information to the figure legend “only the most important significant differences are indicated by astericks”. Please indicate in the fig. legend what R1 and R2 mean.
  14. 4. What Lean_R and Fat-R mean ? Please add this info to the figure legend.
  15. 9. “VMH/DMH for thelean, lean-R and fat-R groups” Please add the space after “the” and in front of “lean”.
  16. 9 “Fourfold and 5-fold decreases in…” please use 4-fold.
  17. 9 “The injection of 5.0 μg/kg BW rbresistin increased (p <0.05) circulating RSTN concen-trations (mean ± SEM) in the fat and lean-R sheep compared to the non treated lean ewes (Figure 5).” Please delate (mean ± SEM) or alternatively provide values.
  18. 9, fig. 5 – legend. Please delate “Values are means (±SEM; ** p < 0.001) , *** p < 0.001 denote difference between groups)” and add “*p<0.05.
  19. 9. Please add spaces in the sentences between words: “Circulating concentrations (mean ± SEM) of leptin after rbresistin treatment were higher(p < 0.01) inthe fat and lean-R sheep than in thelean group.”
  20. 10, figure 6 – legends, please add ** and *** and the description.
  21. 11. “Administration of a relatively low dose of RSTN (1.0 or 5.0 μg per kg BW) sig-nificantly increased the mRNA transcripts of LRa in two experiments in groups of LD sheep (R1 group) fed ad libitum and lean-R sheep nutritionally restricted.” Why Authors chosen above concentrations of RSTN. Please provide rational and cite appropriate literature.
  22. 12 In the sentence: “As mentioned at the beginning of the discussion, we did not observe the detection of LRa in the VMH/DMH in the second experiment or in the ARC, POA and VHM/VDH during the SD season in the first experiment”, please replace or with and
  23. Authors should discuss limitations of their study as only mRNA was examined. What about the protein levels for the short form of the leptin receptors ? Why Authors did nor examine it ? What is known about protein expression for this receptor based on the literature ?

Author Response

Reviewer # 1

We took into account all comments when revising the manuscript.

A detailed list of small changes is presented below, and we indicated changes directly in the text using red font.

Major points:

  1. Please add space between “[4]” and “to” in the sentence: “(ovariectomized with estrogen replacement) as described in detail by Biernat et al. [4]to”

The space is placed.

2.     Please delete “mean ± SD “ after (62 ± 3 kg)

“mean ± SD” is deleted.

  1. Please re-write the following sentence so it is understood “The objective was to alter the BCS of the animals to 2 after Russel et al. [19].”

At the beginning of experiment the average BCS was 3.1 ± 0.3. The aim of the experiment was to obtain a BCS score in animals of level 2 [19].

  1. Authors wrote: “All hormone data are presented as the mean ± SEM. Data analysis was performed by a series of two-way ANOVAs using SigmaPlot® statistical software (version 11.0; Systat Software Inc., Richmond, CA, USA), preceded by Grubb’s test to identify outliers.” Could you please provide information how many outliers you had in this experiments ?

Blood samples (5 mL) were collected at 10-min intervals for 4 hr from each ewe, that means that we had 24 samples from each animals. Each group of animals in the first and the second experiment consisted of 5 animals. So, we run assays in duplicate for each sample and finally we got results from 120 x 2 = 240 samples for each group.

  1. (bottom) “In the AP, during the LD season, the expression of LRa increased proportionally with the administered dose of rbresistin: 2-fold (p < 0.01) for the R1 group and 2.5-fold (p ≥ 0.01)”. Should be p<0.01.

The Reviewer is right. The p value is corrected.

  1. In the SD season, exogenous RSTN decreased (p ≥ 0.001) the mRNA transcript level of LRa in the R1 group and R2 group (Figure 1b)” Should be p<0.001

The Reviewer is right. The p value is corrected.

  1. Should be Figures 1A and B (plural) not “Figure 1A and B”.

The Reviewer is right. We corrected: Figures 1A and B.

  1. Please reorganize the figure so data are presented from the rostral to caudal part of the brain. The data should be presented in the following direction: POA, VMH/DMH, ARC and lastly AP. The same rule applies to Fig. 4.

We agree with Reviewer and presented Figures 1 and 4 in the following direction: POA, VMH/DMH, ARC and lastly AP.

  1. In the sentence “LRa expression was not detected in the ARC or POA during the SD season” please replace “or” with “and”

We agree with Reviewer and replaced “or” with “and”.

  1. “…the LRa transcripts in the POA decreased 3-fold (p <0.001)in the R1 group and 2-fold”, please inset a space between “(p <0.001)” and “in”

The space is placed.

  1. Figure 2 – Please indicate only the most significant and important statistical differences which are disused in the results. Please add this information to the figure legend “only the most important significant differences are indicated by asterisks”.

The changes have been made on Figure 2.

  1. During the LD season, the administration of 1.0 μg/kg BW rbresistin(p < 0.01)”. Please insert a space between “rbresistin” and “(p < 0.01)”.

The space is placed.

  1. Please delete “Means without a common letter differ” from the Fig. 3 legend as there is no letter on this fig. Please indicate only the most significant and important statistical differences which are disused in the results. Please add this information to the figure legend “only the most important significant differences are indicated by astericks”. Please indicate in the fig. legend what R1 and R2 mean.

The sentence from figure 2 caption is deleted. The proper sentence is indicated.

  1. Lean-R and Fat-R mean? Please add this info to the figure legend.

The information is added.

Mean concentrations of circulating (± SEM) of resistin in saline and recombinant bovine resistin-treated groups (R1 – low dose and R2 – high dose rbresistin) during the long- (LD) and the short- (SD) photoperiods.

  1. VMH/DMH for the lean, lean-R and fat-R groups” Please add the space after “the” and in front of “lean”.

Spaces are added.

  1. Fourfold and 5-fold decreases in…” please use 4-fold.

The four-fold is changed for 4-fold.

  1. The injection of 5.0 μg/kg BW rbresistin increased (p <0.05) circulating RSTN concen-trations (mean ± SEM) in the fat and lean-R sheep compared to the non treated lean ewes (Figure 5).” Please delete (mean ± SEM) or alternatively provide values.

The Reviewer is right, we deleted (mean ± SEM), without value that does not sense.

  1. 5 – legend. Please delete “Values are means (±SEM; ** p < 0.001) , *** p < 0.001 denote difference between groups)” and add “*p<0.05.

The sentence is deleted. p value has been changed.

  1. Please add spaces in the sentences between words: “Circulating concentrations (mean ± SEM) of leptin after rbresistin treatment were higher(p < 0.01) inthefat and lean-R sheep than in thelean

Spaces are added.

  1. figure 6 – legends, please add ** and *** and the description.

** p < 0.01, *** p < 0.001. Proper p values are added.

  1. “Administration of a relatively low dose of RSTN (1.0 or 5.0 μg per kg BW) sig-nificantly increased the mRNA transcripts of LRa in two experiments in groups of LD sheep (R1 group) fed ad libitum and lean-R sheep nutritionally restricted.” Why Authors chosen above concentrations of RSTN. Please provide rational and cite appropriate literature.

It is not easy to work on the effects of resistin in sheep, especially when using an "in vivo" model. Our group is the first to administer resistin to large animals intravenously. Previously and nowadays, there is no in vivo work on large domestic animals with resistin dosing in any way. Yes, many studies have been published before, including one work using immunochistochemical localization of resistin in ovine uterus (Dall’Aglio et al., 2019), but most of them are studies using an in vitro model and different species (Spicer et al., 2011; Reverchon et al., 2014; Rak et al., 2015; Sarmento-Cabral et al., 2017).

When I was writing the project included resistin study in 2015, I knew that prof. Arieh Gertler is working on preparation of recombinant ovine resistin. Since 1999, I have been working intensively with the professor and from him I have bought recombinant ovine leptin and its pegylated and non-pegylated antagonists for studies I published for many years. I also went to Rehovot to Institute of Biochemistry, Food Science and Nutrition, Robert H. Smith Faculty of Agriculture, Food and Environment, The Hebrew University of Jerusalem in 2016 to help to develop recombinant resistin. Unfortunately, the professor has so far failed to obtain an active form of this adipokine.

Fortunately, I found paper by Reverchon et al., on cattle (2014) in which he used recombinant bovine resistin purchased in CliniSciences in France, for in vitro study. I bought that rbresistin and started the preliminary experiments.

Before selecting the dose, we conducted a lot of experiments on finding the activity of bovine recombinant resistin in sheep and checked many radio or immuno-tests that would help us to determine the concentration of resistin in sheep. To calculate the dose of resistin for in vivo experiment we based on information which I got from description of Sheep Resistin Elisa kit (BlueGene, Shanghai, China) and preliminary results.

During preliminary experiments, ewes were randomly allocated to one of the three experimental groups (n = 10/group). Treatment groups were as follows: 1) Control, treated with saline (n = 10); 2) R1, treated with a low dose of rbresistin (1.0 µg/kg BW; n = 10); and 3) R2, treated with a high dose of rbresistin (10.0 µg/kg BW; n = 10). Afterwards, concentration of resisin was determined using Sheep Resistin Elisa kit (BlueGene, Shanghai, China) which indicated that basal resistin concentration in Control group was between 3.25-5.3 ng/ml - mean 4.9 ± 0.05 ng/ml, injection of 1.0  µg/kg BW of rbresistin increased (p < 0.05) concentration to 7.5 ± 0.2 ng/ml and the dose of 10.0  µg/kg BW of rbresistin increased (p < 0.01) concentration to 9.2 ± 0.3 ng/ml (unpublished data).

After checking the activity of rbresistin and deciding that it is working perfectly in sheep, we start conducting the studies.

There is small – 8 amino acids difference between bovine and ovine resistin.

Aminoacids sequence homology between ovine and bovine resistin is 91,21%.  Dissimilarities in aminoacid sequences are localized as indicated herein: [10: Val/Ile, 16: Asp/Glu, 22: Leu/Val, 24: Glu/Gly, 28: Asn/Ile, 34: Glu/Arg, 53: Ser/Gly, 89: Arg/His].

Green – ovine resistin

Black– bovine resistin

QSLCPIDKAVSKKIQDVTTSLLPEAVRNIGLDCQSVTSRGSLVTCPSGFAVTSCTCGSAC

QSLCPIDKAISEKIQEVTTSLVPGAVRIIGLDCRSVTSRGSLVTCPSGFAVTGCTCGSAC

GSWDVRAETTCHCQCAGMDWTGARCCRLRIQ

GSWDVRAETTCHCQCAGMDWTGARCCRLHIQ

For the first set of experiments we decided to use doses of 1 and 10 µg/kg BW. Having worked on leptin for 20 years, we knew that very low doses of leptin are more effective in ruminants than higher doses. Hence the choice of these two different doses. We published first data in Domestic Animal Endocrinology in 2018.

When starting further experiments, due to the very high price of 1 mg of rbresistin, we could no longer use 2 doses, hence we decided to take an intermediate dose of 5 µg/kg BW.

For the first experiment we used for 1 ewe: 60 kg x 1 µg/kg = 60 µg/kg x 15 ewes = 900 µg

                                                                     60 kg  x 10 µg/kg = 600 µg/kg x 15 ewes = 9 mg

Cat# Price Qty Total Excl. VAT NB-CUST-CAT-16032017-1

Recombinant Bovine Resistin produced in E. Coli     11 040,00 € -15%= 9 384,00 €  for 10 mg

The second experiment was completely new for our group. In that experiment we work on two models: lean and fat ewes; so we think that the decision about 5 µg/kg BW was right. Above, you can find catalogue price for rbresistin with 15% discount I got.

This decision also resulted from the fact that we allocated a much smaller amount for the purchase of the resistin, hoping to buy it from prof. Gertler. Since we didn't make it, we couldn't afford to use further the dose of 10 µg/kg.

Dall’Aglio, C.; Scocco, P.; Maranesi, M.; Petrucci, L.; Acuti, G.; De Felice, E.; Mercati, F. Immunohistochemical identification of resistin in the uterus of ewes subjected to dfferent diets: Preliminary results. Eur. J. Histochem. 2019, 63, 3020.

Spicer, L.J.; Schreiber, N.B.; Lagaly, D.V.; Aad, P.Y.; Douthit, L.B.; Grado-Ahuir, J.A. Effect of resistin on granulosa and theca cell function in cattle. Anim. Reprod. Sci. 2011, 124, 19–27.

Reverchon, M.; Ramé, C.; Cognié, J.; Briant, E.; Elis, S.; Guillaume, D.; Dupont, J. Resistin in dairy cows: Plasma concentrations during early lactation, expression and potential role in adipose tissue. PLoS One 2014, 9, e93198; DOI:10.1371/journal.pone.0093198

Rak, A.; Drwal, E.; Karpeta, A.; Gregoraszczuk, E.L. Regulatory role of gonadotropins and local factors

produced by ovarian follicles on in vitro resistin expression and action on porcine follicular  steroidogenesis. Biol. Reprod. 2015, 92, 142.

Sarmento-Cabral A, Peinado JR, Halliday LC, Malagon MM, Castaño JP, Kineman RD, Luque RM. Adipokines (leptin, adiponectin, resistin) differentially regulate all hormonal cell types in primary anterior pituitary  cell cultures from two primate species. Sci Rep 2017;7:43537.

Biernat W., Szczesna M., Kirsz K., Zieba D.A. Resistin regulates reproductive hormone secretion from the ovine adenohypophysis depending on season. Domest. Anim. Endocrinol. 2018:65:95-100,10.1016/j.domaniend.2018.

  1. In the sentence: “As mentioned at the beginning of the discussion, we did not observe the detection of LRa in the VMH/DMH in the second experiment orin the ARC, POA and VHM/VDH during the SD season in the first experiment”, please replace or with and

The replacement has been done.

  1. Authors should discuss limitations of their study as only mRNA was examined. What about the protein levels for the short form of the leptin receptors ? Why Authors did not examine it ? What is known about protein expression for this receptor based on the literature ?

The experiments we presented in manuscript are part of grant project in which we want to determine expression of LRa on mRNA level. When deciding on the methodology of experiments and determination of expression of LRa at the mRNA level, we took into account the work in which the Authors mentioned that: “Determination of ObR subtypes at the protein level is still hampered by the lack of ObRa-, ObRc-, and ObRd-specific antibodies, although ObRb and ObRe may be differentiated by their different lengths on Western blot. By far, real-time PCR is the most sensitive method to quantify the differences among the isoforms.”

Pan W., Hsuchou H., Tu H., Kastin A.J. Developmental Changes of Leptin Receptors in Cerebral Microvessels: Unexpected Relation to Leptin Transport. Endocrinology 2008,  149(3):877– 885.

There is no information in literature about LRa protein expression.

Furthermore, as I mentioned above due to few circumstances as high price of resistin and we had no more money to try to determine protein expression.

Reviewer 2 Report

This is an interesting study that gives information on seasonal and nutritional fluctuations in the mRNA Levels of the short form of the Leptin Receptor (LRa) in the hypothalamus and anterior pituitary in resistin-treated sheep. Short form of leptin receptor expression levels is affected by day length, adiposity and resisting in sheep. Also, the manuscript suffers a lack of clarity. The presentation needs several improvements.

  1. It is noted that your manuscript needs careful editing paying particular attention to English spelling, and sentence structure so that the goals and results of the study are clear to the reader, especially in 3.2.1 part.
  2. Figures 2 and 3 are grouped by LD and SD to make it clearer, like Figure 1.
  3. In 3.2 part, why is LD marked in Figure 4, 5 and 6?

Author Response

Reviewer # 2

We took into account all comments when revising the manuscript.

A detailed list of small changes is presented below, and we indicated changes directly in the text using red font.

The manuscript suffers a lack of clarity. The presentation needs several improvements.

  1. It is noted that your manuscript needs careful editing paying particular attention to English spelling, and sentence structure so that the goals and results of the study are clear to the reader, especially in 3.2.1 part.

The English has been checked by professionalists from American Journal Expert. I have been working with those company for years and the paper is now double-checked for spelling and I hope that Reviewer is satisfied now. I add the proper certificate as an attachment.

  1. Figures 2 and 3 are grouped by LD and SD to make it clearer, like Figure 1.

  1. In 3.2 part, why is LD marked in Figure 4, 5 and 6?

There are two experiments in the study concerning the effects of few factors engaged in leptin resistance in sheep as a physiological adaptation: resistin, lower transfer of leptin through BBB using LRa and adiposity. Sheep as seasonal animals are under the strong influence of photoperiod why we carried out the first experiment during the long (LD) and short (SD) photoperiods.

Natural leptin resistance only occurs in sheep during a long day (LD), after conducting a study (experiment 1) in which we examined the effect of photoperiod as an experimental factor, we decided to investigate in a second experiment how changes in body weight and resistin affect the transfer of leptin to the brain to use LRa only when the days get longer.

In the second experiment an alternation of body weight that we performed over the course of 5 months allowed us to obtain a lean and fat sheep model.

To sum up: four mechanisms for leptin resistance have been proposed in literature: 1) a decrease in leptin transport (using LRa) from the bloodstream to the hypothalamus (termed peripheral resistance), 2) desensitization of leptin receptors (termed central resistance, 3) suppression of the long isoform of the leptin receptor (LRb)-associated signaling pathways by negative regulators like suppressor of cytokine signaling (SOCS)3 (also termed central resistance), and 4) hypothalamic inflammation, which alters the neuronal pathway involved in energy homeostasis.

Since 2005 we recognized that SOCS3 is involved in creation of hypothalamic leptin insensitivity:

D.A. Zieba, M. Szczesna, B. Klocek-Gorka, E. Molik, T. Misztal, G.L. Williams, K. Romanowicz, E. Stepien, D.H. Keisler, M. Murawski. Seasonal effects of central leptin infusion on melatonin and prolactin secretion and on SOCS-3 gene expression in ewes. J. Endocrinol. 2008; 198: 147-155.

              D.A. Zieba, M. Szczesna, B. Klocek-Gorka, G.L. Williams. Leptin as a nutritional signal regulating an                                                   appetite and reproductive processes in seasonally- breeding ruminants. J. Physiol.Pahrmacol. 2008; 59 (9): 7-18.

  1. Szczesna, D.A. Zieba, B. Klocek-Górka, T. Misztal, E., Stepien. Seasonal effects of central leptin infusion and prolactin treatment on pituitary SOCS-3 gene expression in ewes. J. Endocrinol. 2011; 208: 81-88.

Szczesna M, D.A. Zięba. Phenomenon of leptin resistance in seasonal animals: the failure of leptin action in the brain. Domest. Anim. Endocrinol. 2015; 52:60-70.

Later, we investigated the insensitivity/resistance to leptin that occurs naturally during pregnancy:

Szczesna M., Kirsz K., Kmiotek M., ZiÄ™ba D.A. Seasonal fluctuations in the steady-state mRNA levels of  suppressor of cytokine signaling-3 (SOCS-3) in the mammary gland of lactating and non-lactating ewes. Small Rumin. Res. 2015;124: 101-104

Szczesna M., Kirsz K., Misztal T., Zieba D.A. Downregulation of LRb and upregulation of SOCS-3 during pregnancy in sheep - implications for leptin resistance Downregulation of LRb in VMH/DMH during the second half of gestation and upregulation of SOCS-3 in ARC in late-pregnant ewes - implications for leptin resistance. Gen. Comp. Endocrinol. 2019:274:73-79

This manuscript is the fifth in which we draw attention to the role of resistin in the process of leptin resistance and to a decrease in leptin transport using LRa from the bloodstream to the hypothalamus (termed peripheral resistance).

Zieba D.A., Biernat W., Szczesna M., Kirsz K., Misztal M. Hypothalamic-pituitary and adipose tissue responses to the effect of resistin in sheep: integration of leptin and resistin signaling involving suppressor of cytokine signaling 3 and the long form of the leptin receptor. Nutrients 2019:11,2180.

Zięba D.A. Factors affecting the transfer of leptin through blood-brain barrier (BBB). The implication for leptin resistance. Appro. Poult. Dairy & Vet. Sci. 2019: 6 (5):595-598

Zieba D.A., Biernat. W. Barć J. The roles of leptin and resistin in reproduction and leptin resistance in sheep. Domest. Anim. Endocrinol. 2020: 73:1.

Zieba D.A., Biernat W., Szczesna M., Kirsz K., Barć J, Misztal T. Changes in expression of the genes for the leptin signaling in hypothalamic-pituitary selected areas and endocrine responses to long-term manipulation in body weight and resistin in ewes. Int. J. Mol. Sci. 2020; 21: 4238.

To clarify our goals we added few sentence to Introduction:

Considering the aforementioned factors, the research was divided into two experiments; since sheep are seasonal animals, in the first experiment, we investigated the effect of the length of the day, the photoperiod, and the research was carried out during the long and short photoperiods. The second experiment was conducted during the LD period and highlighted the role of adiposity in the process of leptin resistance, since sheep are leptin resistant during the long day photoperiod. These factors were studied in the context of the expression of the short form of the leptin receptor,LRa, which is responsible for the transport of this adipokine to the brain.

Furthermore in Discussion the sentences are placed:

In our studies, both animal models – LD sheep as a physiological example of leptin resistance and sheep with high BW and high adiposity resulting from diet-induced obesity – demonstrated high leptin concentrations. In those two models, highly elevated expression of LRa was observed in the ARC nuclei – the neurons that transmit the leptin signal first directly to the CNS.

Reviewer 3 Report

This manuscript aims to determine the mRNA expression levels of the short form of the leptin receptor (LRa) in the hypothalamus and anterior pituitary (AP) of a total of fifty female Polish Longwool sheep. The experiment design is reasonable, the experiment plan is rigorous, the experiment result can explain and confirm the scientific problem. For the purpose of this paper, the following suggestions are made:

  1. Whether the effects of male and female, age and breed of sheep were considered in the experiment design.
  2. The physiological leptin concentration in LD season was higher than that in SD season, but the metabolic mechanism between physiological leptin concentration and season was not clarified.
  3. The background aboutthe long isomer LRB in the introduction is quite lengthy.
  4. The 5thparagraph in discussion can be shortened as the discussion and should focus on the results described.

Author Response

Reviewer # 3

We took into account all comments when revising the manuscript.

A detailed list of small changes is presented below, and we indicated changes directly in the text using red font.

  1. Whether the effects of male and female, age and breed of sheep were considered in the experiment design.

There are two experiments in the study concerning the effects of few factors engaged in leptin resistance in sheep as a physiological adaptation: resistin, lower transfer of leptin through BBB using LRa and adiposity. Sheep as seasonal animals are under the strong influence of photoperiod why we carried out the first experiment during the long (LD) and short (SD) photoperiods.

Natural leptin resistance only occurs in sheep during a long day (LD), after conducting a study (experiment 1) in which we examined the effect of photoperiod as an experimental factor, we decided to investigate in a second experiment how changes in body weight and resistin affect the transfer of leptin to the brain to use LRa only when the days get longer.

In the second experiment an alternation of body weight that we performed over the course of 5 months allowed us to obtain a lean and fat sheep model.

Yes, the age was taking into account: only ewes between 2 and 3 yrs of aged were intended for research, after one or two pregnancies and ovarectomy was performed to unify steroidogenic status.

Yes, the breed was the most important because Polish longwool sheep is a breed that shows strong breeding seasonality and that breed we used in all our experiments concerning leptin, resistin or other  orexigenic or anoxigenic peptides studies. Main goal of our experiments is to recognized how metabolic factors influence seasonality phenomenon and leptin resistance.

  1. The physiological leptin concentration in LD season was higher than that in SD season, but the metabolic mechanism between physiological leptin concentration and season was not clarified.

In experiment 2 we have shown that indeed resistin is able to increase leptin concentrations as was indicated in rats by Asterholm et al. (2014) and it is another factor engaged in hyperleptinemia in sheep during LD.  However, our previous study published in International Journal of Molecular Sciences form 2020 Zieba D.A., Biernat W., Szczesna M., Kirsz K., Barć J, Misztal T. Changes in expression of the genes for the leptin signaling in hypothalamic-pituitary selected areas and endocrine responses to long-term manipulation in body weight and resistin in ewes, described more metabolic parameters concerning hyperleptinemia during LD in sheep.

Furthermore, our other experiments concerned mainly the physiological concentration of leptin depending on the season and some metabolic factors.

D.A. Zieba, B. Klocek, G.L. Williams, K. Romanowicz, L. Boliglowa, M. Wozniak. In vitro evidence that leptin suppresses melatonin secretion during long days and stimulates its secretion during short days in seasonal breeding ewes. Domest. Anim. Endocrinol. 2007; 33(3): 358-365.

  1. Klocek-Górka, SzczÄ™sna M., E. Molik and D.A. ZiÄ™ba. The interactions of season, leptin and melatonin with thyroid hormone secretion, using an in vitro approach. Small Rumin. Res. 2010, 91: 231-235

  1. Szczęsna, D.A. Zieba-Przybylska, B. Klocek-Gorka. D.H. Keisler. Interactive effects of prolactin, growth hormone, melatonin and time of year on leptin secretion in ovine adipose tissue: An in vitro study. Small Rumin. Res. 2011; 100: 177-183.

Szczesna M.,  K. Kirsz, M. Kucharski, P. Szymaszek and Zieba D.A..  The seasonal interactions  between leptin and GH and its effect on pituitary SOCS-3 gene expression in sheep. Health. 2013; 5 (8A3): 29-39.

Zieba D.A., Biernat. W. Barć J. The roles of leptin and resistin in reproduction and leptin resistance in sheep. Domest. Anim. Endocrinol. 2020: 73:106472.

Zieba D.A., Biernat W., Szczesna M., Kirsz K., Barć J, Misztal T. Changes in expression of the genes for the leptin signaling in hypothalamic-pituitary selected areas and endocrine responses to long-term manipulation in body weight and resistin in ewes. Int. J. Mol. Sci. 2020, 21, (12): 4238

  1. The background about the long isomer LRB in the introduction is quite lengthy.

As per Reviewer request we delete some sentences from Introduction.

The signal is transmitted via activation of Janus tyrosine kinase 2 (JAK2) by the leptin receptor, which induces the autophosphorylation of this kinase, followed by the phosphorylation of LRb by JAK2 [8]. JAK2 kinase - in complex with LRb - then phosphorylates the STAT3 protein (signal transducer and activator of transcription 3), which is a cytosolic signal transducer and activator of transcription [9].

  1. The 5thparagraph in discussion can be shortened as the discussion and should focus on the results described.

The paragraph 5th in Discussion is shortened.

This phenomenon, also described in those studies, has been known for two decades; compared to the SD season, in the LD season

During this period, when readily available food is plentiful, sheep show an increased appetite and appear to be resistant to the high concentrations of leptin resulting from increased body mass and adiposity.

Round 2

Reviewer 1 Report

Authors responded to my questions and comments. The quality of the manuscript has been improved. I have no more questions.